# Relative Permeability: A Critical Parameter in Numerical Simulations of Multiphase Flow in Porous Media

Nathan Moodie [1], William Ampomah [2], Wei Jia [1] and Brian McPherson [1,*]

1   Carbon Science & Engineering Research Group, University of Utah, Salt Lake City, UT 84112, USA; nathan.moodie@m.cc.utah.edu (N.M.); wei.jia@utah.edu (W.J.)
2   New Mexico Tech, Socorro, NM 87801, USA; william.ampomah@nmt.edu
*   Correspondence: b.j.mcpherson@utah.edu; Tel.: +1-801-558-4043

**Abstract:** Effective multiphase flow and transport simulations are a critical tool for screening, selection, and operation of geological $CO_2$ storage sites. The relative permeability curve assumed for these simulations can introduce a large source of uncertainty. It significantly impacts forecasts of all aspects of the reservoir simulation, from $CO_2$ trapping efficiency and phase behavior to volumes of oil, water, and gas produced. Careful consideration must be given to this relationship, so a primary goal of this study is to evaluate the impacts on $CO_2$-EOR model forecasts of a wide range of relevant relative permeability curves, from near linear to highly curved. The Farnsworth Unit (FWU) is an active $CO_2$-EOR operation in the Texas Panhandle and the location of our study site. The Morrow 'B' Sandstone, a clastic formation composed of medium to coarse sands, is the target storage formation. Results indicate that uncertainty in the relative permeability curve can impart a significant impact on model predictions. Therefore, selecting an appropriate relative permeability curve for the reservoir of interest is critical for $CO_2$-EOR model design. If measured laboratory relative permeability data are not available, it must be considered as a significant source of uncertainty.

**Keywords:** relative permeability; geologic carbon storage; multi-phase flow simulation

## 1. Introduction

With the prospects of climate change looming and an ever-increasing demand for power generation and heavy industry, reducing greenhouse gas emissions from large point-source emitters, such as coal and natural gas power plants or fertilizer operations, has become paramount. Geologic carbon storage (GCS) is one potential path for emission reductions. Carbon dioxide is captured from the large point-source emitters, compressed into a supercritical state and injected into a suitable storage formation such as a depleted oil and gas reservoir or a deep saline aquifer [1–5]. Mature oil fields undergoing $CO_2$-Enhanced Oil Recovery ($CO_2$-EOR) are another promising option for GCS that also offer an economic benefit. The incremental recovery for $CO_2$-EOR operations can produce an additional 7–23% of the oil in place while simultaneously storing roughly 40% of the $CO_2$ injected [6].

Multiphase flow and transport simulation that can characterize $CO_2$ effects in oil and water are an integral part of designing GCS projects for oil and gas fields. These simulations are used in project design, permitting, forecasting oil production and storage capacity, and quantifying possible site risks. Understanding the uncertainty in the simulation model inputs and their impact on performance and predictions is critical for project success. The permeability distribution in a reservoir is probably the biggest source of uncertainty, followed closely by relative permeability. Three-phase relative permeability has significant impacts on fluid flow and storage capacity, yet is poorly understood and often generalized. Laboratory measurements have historically been focused on measuring two- and three-phase relative permeability curves for oil and gas ($CH_4$) reservoirs [7–9]. From these data, empirical models have been developed which promise broad applicability, including use

in GCS numerical simulations [7–14]. For this study, we utilized an empirical formula for relative permeability developed by Corey [10] to described oil and gas flow in porous media. This empirical formula was to create relative permeability curves for $CO_2$-EOR numerical simulations at our field site [15,16].

It is important to understand that when measuring three-phase relative permeability (gas/oil/water), experimental methods generally fall into one of two categories. The first general method measures pairs' two-phase relative permeability, gas/oil and oil/water, and then uses an empirical combination model such as Stone II or the Baker model to calculate the three-phase relative permeability [7,13,14,17]. In the second general method, all three phases are measured concurrently to create a true three-phase relationship for the fluids of interest [8,17,18]. Generally, numerical simulation codes do not leverage three-phase relative permeability data. Simulators such as STOMP use a two-phase empirical model along with critical parameters like the residual wetting and non-wetting phase saturations and a curve parameter to calculate the relative permeability from the fluid saturations [19]. Alternatively, other numerical simulators, such as the Eclipse® numerical simulator used in this study, leverage table data in the form of two-phase saturation versus relative permeability tabular data coupled with linear interpolation between data points [20]. These methods then use a combination model to calculate the three-phase relative permeability. Either method requires the input data to be in the form of a pair of two-phase relative permeability versus saturation curves, one for the gas/oil pair and one for the oil/water pair. In this study, we leveraged the functionality in Eclipse® to use tabular data by generating a suite of plausible curves using the Corey's Curve empirical formula, then importing those curves into the simulation model.

The choice of relative permeability curve can be a significant source of model uncertainty. This uncertainty can come from the uncertainty inherent in laboratory measurements or the general lack of relative permeability curves for most GCS candidate sites and formations. At the time of this study, the target reservoir at our study site had a single pair of binary relative permeability curves that were derived from a laboratory study [21,22]. Studies quantifying the model uncertainty related to the relative permeability curve on numerical simulation forecasts are rare, and for our study site do not exist. Therefore, a primary goal of this study is to evaluate how uncertainty in the relative permeability curves impacts $CO_2$-EOR model forecasts.

## 2. Study Site: Farnsworth Unit, Texas

The study site is the Farnsworth Unit, an active $CO_2$-EOR site since 2010 located in the Anadarko Basin of northern Texas. The target formation is in the Upper Morrow sequence called the Morrow 'B' Sandstone. The field has produced 27 billion cubic feet of gas and more than 1.9 million barrels of oil from the Morrow 'B' Sandstone, a fluvial valley-fill sandstone [22–24]. The formation is at a depth of between 7550 and 7950 feet and is a series of connected sandstone bodies dipping to the West at less than one degree [21,23,25]. The sandstone sequences occur at the base of the 'B' unit and are up to 44 ft thick consisting of medium to coarse sand and conglomerates that are thought to be deposited as a series of fluvial point-bars that are all connected [23,24,26]. Laboratory measurements on core samples and well log analysis indicate a mean porosity of 14% to 17% and average permeabilities that range from 27 mD to 140 mD [24,25,27] The older work done by Bolyard (1989) indicated higher porosity and permeability while the newer work done by Rose-Coss et al. (2016) indicated lower average porosity and permeability [24,26].

## 3. Numerical Model Development

### 3.1. FWU Geological Model

A geological model of the Farnsworth Unit was developed by the Southwest Regional Partnership on Carbon Sequestration (SWP) using the full suite of petrophysical data collected during the ongoing characterization effort including SEM, XRD, seismic data sets, well logs, core samples, and thin sections [26–29]. The full geological model represents

the whole oil field at Farnsworth. Petrophysical properties of porosity and permeability were populated across the domain by stochastic algorithm constrained by well logs [16]. A subset of the domain centered on the west half of the field was used for this study. This is the current active injection and production area and primary focus of the SWP research.

### 3.2. Relative Permeability

For this study, 17 different relative permeability curves were constructed that represent a possible set of curves which may apply to the reservoir. The goal is to study a wide range of relative permeability curves with the assumption that as the relative permeability curve becomes more linear, the saturation end-points and the relative permeability end-points become larger, representing a transition from low fluid mobility to high fluid mobility. The residual phase saturation, maximum relative permeability, and the curve shape were varied to create a range of curves that bracket the parameter space from near linear to highly curved. As of the time of this study, there was only a single relative permeability curve measured for the Morrow Sandstone [21,22]. This lack of measured data necessitates that a wide range of input parameters be examined to understand the influence relative permeability has on a $CO_2$-EOR operation. Figure 1 highlights a representative selection of curves that transition from highly curved to near linear. The remaining relative permeability curves used in the study are shown in Figures 2, A1 and A2.

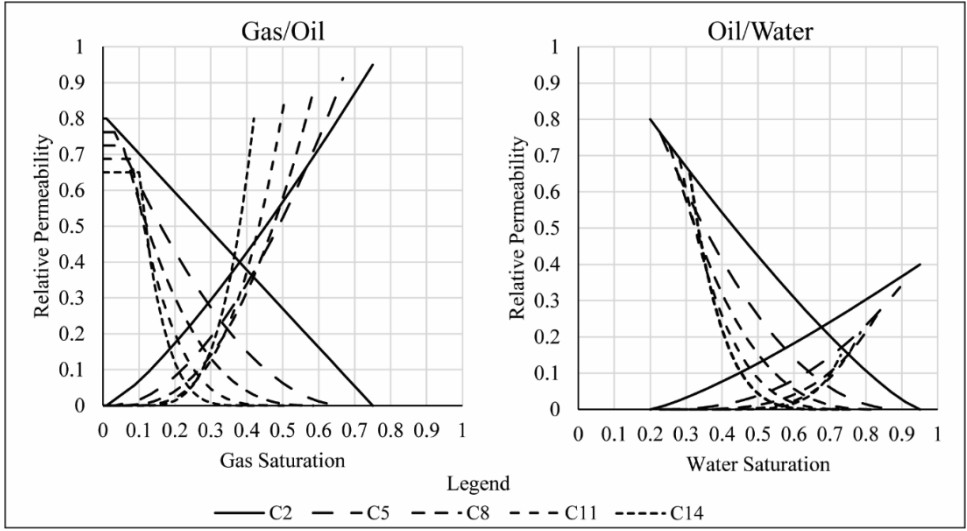

**Figure 1.** A representative selection of relative permeability curves used in this study. All curves used follow a similar trend of high residual saturation and highly curved to low residual saturation and near linear curve. Only those shown here vary the maximum relative permeability in addition to the other two variables.

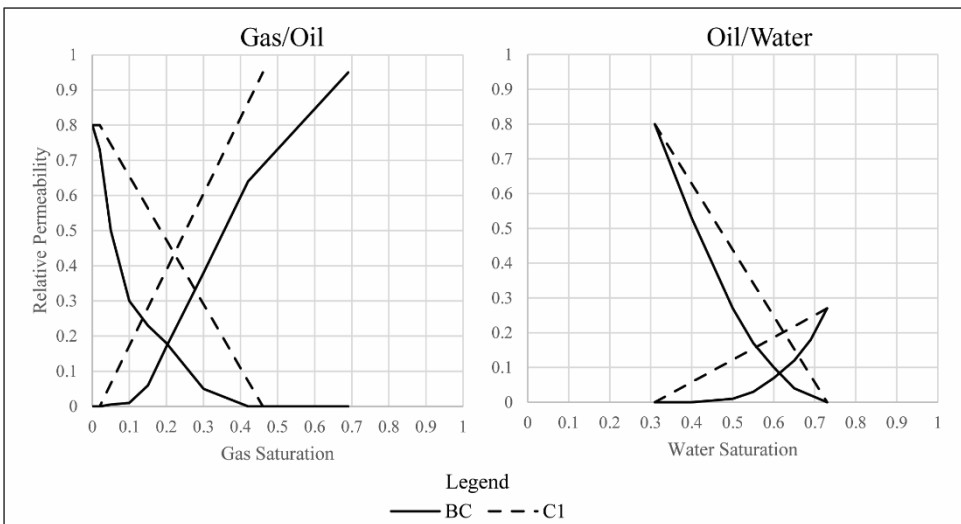

**Figure 2.** Plot shows the Base Case relative permeability curve from a UNOCAL study (solid lines) and the C1 relative permeability curve that represents the most linear curve used in the study.

In this study we used the Eclipse® numerical simulation software for all the simulations. A key feature that we leveraged is its ability to use lookup tables with linear interpolation between points instead of empirical formulas to calculate the relative permeability. This effectively decouples the empirical formula from the numerical simulator, allowing much greater flexibility in specifying the saturation and relative permeability endpoints as well as the degree of curvature. Any empirical formula could have been chosen to create the curves.

We chose to use a modified version of the Corey's Curve formula to create all the Eclipse® lookup tables used except for the Base Case, discussed below. This relationship provides great flexibility for developing a wide variety of curves from highly non-linear to linear with the added ability to modify the residual phase saturation and maximum relative permeability endpoints. The Corey's Curve formula allows the creation of a wide range of relative permeability curves based on three sets of inputs, residual saturation, maximum relative permeability, and lambda (the curve parameter). The gas and non-aqueous liquid (oil) curve are described by Equations (1) and (2), where $S_t$ is the liquid saturation defined as the non-aqueous liquid (oil) saturation plus the residual aqueous liquid saturation, $S_{gr}$ is the residual gas saturation, $S_{tr}$ is the residual liquid saturation defined as the residual oil saturation plus the residual water saturation, $\lambda$ defines the shape of the curve, $k_{rg}max$ is the maximum gas relative permeability, $k_{rn}max$ is the maximum non-aqueous liquid (oil) relative permeability, and $k_{rg}(S_t)$ and $k_{rn}(S_t)$ are the gas and non-aqueous liquid relative permeability at liquid saturation $S_t$ [19].

$$k_{rg}(S_t) = k_{rg}max(1 - (S_t - S_{tr})/(1 - S_{tr} - S_{gr}))^\lambda \qquad (1)$$

$$k_{rn}(S_t) = k_{rn}max((S_t - S_{tr})/(1 - S_{tr} - S_{gr}))^\lambda \qquad (2)$$

The water and oil curves are described by Equations (3) and (4), where $S_l$ is the aqueous saturation, $S_{lr}$ is the residual aqueous saturation, $S_{nr}$ is the residual non-aqueous (oil) saturation, $\lambda$ defines the shape of the curve, $k_{rl}max$ is the maximum aqueous liquid relative permeability, $k_{rn}max$ is the maximum non-aqueous liquid relative permeability, and $k_{rl}(S_l)$ and $k_{rn}(S_l)$ are the aqueous and non-aqueous liquid relative permeability at aqueous liquid saturation $S_l$ [19]. Tables 1 and 2 have the parameters (residual phase saturation, maximum phase relative permeability, and lambda) used to calculate all the relative permeability curves in this study, including the subset shown in Figure 1.

$$k_{rl}(S_l) = k_{rl}max((S_l - S_{lr})/(1 - S_{lr} - S_{nr}))^\lambda \qquad (3)$$

$$k_{rn}(S_l) = k_{rn}max(1 - (S_l - S_{lr})/(1 - S_{lr} - S_{nr}))^{\lambda} \tag{4}$$

**Table 1.** Phase residual saturation and curve parameter lambda used in Equations (1)–(4) to create the relative permeability curves used in this study.

| Model | $S_{gr}$ | $S_{tr}$ | $S_{lr}$ | $S_{nr}$ | Lambda(G/O) Oil | Gas | Lambda(O/W) Oil | Water |
|-------|----------|----------|----------|----------|-----|-----|-----|-------|
| C1 | 0.020 | 0.540 | 0.310 | 0.270 | 1.000 | 1.250 | 1.250 | 1.250 |
| C2 | 0.01 | 0.25 | 0.2 | 0.05 | 1 | 1.25 | 1.25 | 1.25 |
| C3 | 0.01 | 0.25 | 0.2 | 0.05 | 1 | 1.25 | 1.25 | 1.25 |
| C4 | 0.07 | 0.2 | 0.2175 | 0.0875 | 1 | 1.25 | 1.25 | 1.25 |
| C5 | 0.0325 | 0.3325 | 0.2275 | 0.105 | 1.875 | 1.8125 | 2.0625 | 2.4375 |
| C6 | 0.0325 | 0.3325 | 0.2275 | 0.105 | 1.875 | 1.8125 | 2.0625 | 2.4375 |
| C7 | 0.13 | 0.1433 | 0.2533 | 0.12 | 1.875 | 1.8125 | 2.0625 | 2.4375 |
| C8 | 0.055 | 0.415 | 0.255 | 0.16 | 2.75 | 2.375 | 2.875 | 3.625 |
| C9 | 0.055 | 0.415 | 0.255 | 0.16 | 2.75 | 2.375 | 2.875 | 3.625 |
| C10 | 0.08 | 0.1 | 0.15 | 0.09 | 2.75 | 2.375 | 2.875 | 3.625 |
| C11 | 0.0775 | 0.4975 | 0.2825 | 0.215 | 3.625 | 2.9375 | 3.6875 | 4.8125 |
| C12 | 0.0775 | 0.4975 | 0.2825 | 0.215 | 3.625 | 2.9375 | 3.6875 | 4.8125 |
| C13 | 0.0733 | 0.26 | 0.3 | 0.153 | 3.625 | 2.9375 | 3.6875 | 4.8125 |
| C14 | 0.1 | 0.58 | 0.31 | 0.27 | 4.5 | 3.5 | 4.5 | 6 |
| C15 | 0.1 | 0.58 | 0.31 | 0.27 | 4.5 | 3.5 | 4.5 | 6 |
| C16 | 0.12 | 0.13 | 0.215 | 0.15 | 4.5 | 3.5 | 4.5 | 6 |

**Table 2.** Maximum relative permeability values used in Equations (1)–(4) to create the relative permeability curves used in this study.

| Model | $k_{rn}(S_t)Max$ | $k_{rg}(S_t)Max$ | $k_{rl}(S_l)Max$ | $k_{rn}(S_l)Max$ |
|-------|------------------|------------------|------------------|------------------|
| C1 | 0.8 | 0.95 | 0.27 | 0.8 |
| C2 | 0.8 | 0.95 | 0.4 | 0.8 |
| C3 | 0.8 | 0.95 | 0.27 | 0.8 |
| C4 | 0.8 | 0.95 | 0.27 | 0.8 |
| C5 | 0.7625 | 0.9125 | 0.3375 | 0.7625 |
| C6 | 0.8 | 0.95 | 0.27 | 0.8 |
| C7 | 0.8 | 0.95 | 0.27 | 0.8 |
| C8 | 0.725 | 0.875 | 0.275 | 0.725 |
| C9 | 0.8 | 0.95 | 0.27 | 0.8 |
| C10 | 0.8 | 0.95 | 0.27 | 0.8 |
| C11 | 0.6875 | 0.8375 | 0.2125 | 0.6875 |
| C12 | 0.8 | 0.95 | 0.27 | 0.8 |
| C13 | 0.8 | 0.95 | 0.27 | 0.8 |
| C14 | 0.65 | 0.8 | 0.15 | 0.65 |
| C15 | 0.8 | 0.95 | 0.27 | 0.8 |
| C16 | 0.8 | 0.95 | 0.27 | 0.8 |

The Base Case model permutation uses a relative permeability curve (BC in Figure 2) developed by UNOCAL for a simulation study focused on the efficacy of water and $CO_2$ flooding at the Farnsworth Unit [21,22]. The UNOCAL study did not provide a capillary pressure curve for the Morrow 'B' Sandstone. For this site, it is initially believed that capillary pressure had a negligible effect on phase movement and was thus ignored in the previous models to aid in simplicity and computational speed. For consistency with previous work and to reduce extra variables that may influence the simulation results capillary pressure was not included [16,25]. We plan to study the influence that the addition of capillary pressure may have on model forecasts to determine if this assumption is valid.

### 3.3. Farnsworth Units Model Domain

The Farnsworth Unit is divided into an east and west half that appear to be hydraulically split [22,23]. The west half is the site of most production and injection operations historically and are where current and future $CO_2$-EOR operations are occurring [22,23]. A detailed geological model encompassing the Morrow 'B' Sandstone in the west half of the reservoir was the basis of our simulation model [26]. We up-scaled this geologic model with over 26 million cells to a simulation model consisting of 33,756 active cells. Along with the grid geometry, we up-scaled the permeability and the porosity (Figure 3). This yielded a mean permeability of 39 mD with a standard deviation of 54 mD and a mean porosity of 14% with a standard deviation of 3%, in line with the original geological model values of 13.6% porosity and 27 mD permeability. We assumed the sealing formation, the Morrow Shale, made a no-flow boundary on all sides as well as the top and bottom of the reservoir, so only the reservoir interval is included in the simulation model. See Moodie et al. (2019) for more details on the model domain description [30].

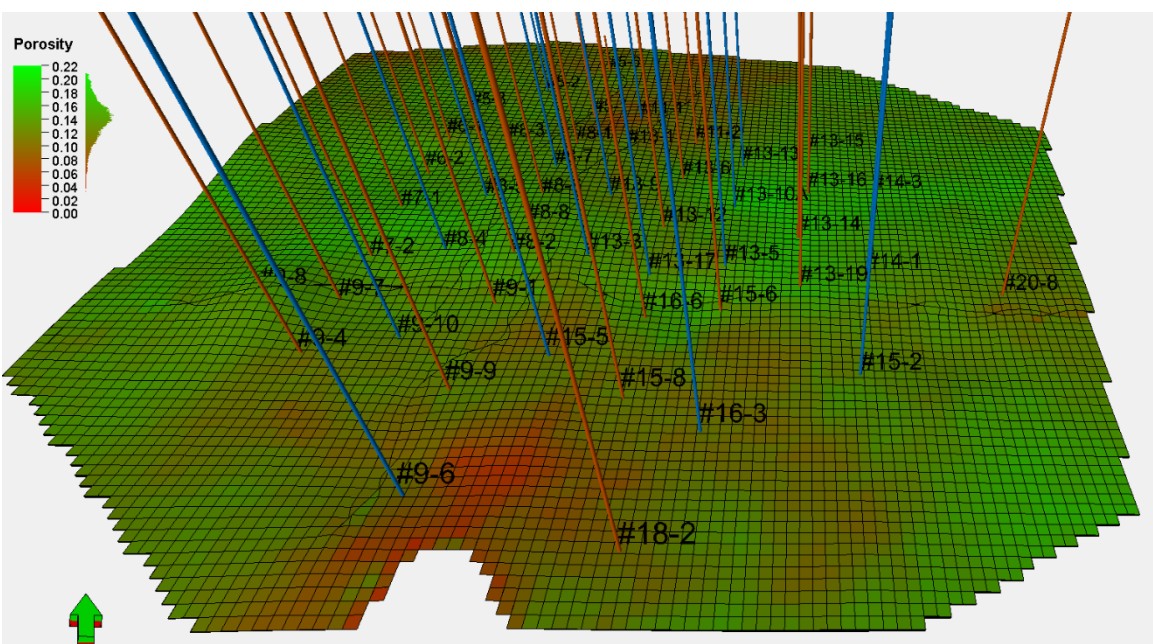

**Figure 3.** The dynamic model domain showing the porosity distribution for the top of the model with the injection wells in blue and production wells in red.

The initial conditions were derived from the results of the history-matched primary production and water-flooding modeling study done previously by the SWP and include the oil saturation and oil component distribution, the water saturation and the pressure [16]. The simulation is initialized with no gas phase. All methane ($CH_4$) and other light volatiles are dissolved in the oil. A compositional fluid model is specified for this study based on a fluid properties report from initial exploration, then refined by fluid modeling [31,32].

### 3.4. Well Operations Schedule and Model Fit

The Farnsworth Unit operates under a water alternating gas (WAG) injection scheme. The schedule used in this study mimics current practices and future plans. Figure 3 indicates the location of the injection and production wells used in the model. The production and injection schedule are broken into two main operation periods. The first period extends from 1 December 2010 to 31 July 2016 and uses historical monthly injection and production data to define well rates. The second period extends from 1 August 2016 to 1 January 2036 and models potential future operations through to the end of the field's lifetime. During this second period, the $CO_2$ injection volume gradually decreases until it is completely reliant on recycled $CO_2$ to meet targets. See Moodie et al. (2019) for a detailed breakdown of the operations schedule [30].

To assess the model performance, we perform a regression analysis comparing the historical oil production from December 2010 to January 2016 to the Base Case simulation model. A $R^2$ value of 0.94 indicates a reasonable fit and comparing this data to a history-matched model developed by Ampomah et al. (2016) [16]. Reviewing Figure 4 indicates a strong correlation between the historical data and the model data, with the Base Case (dashed line), the history-matched model of Ampomah et al. (2016) (solid black line), and the FWU historical data (open circles) all plotted [16].

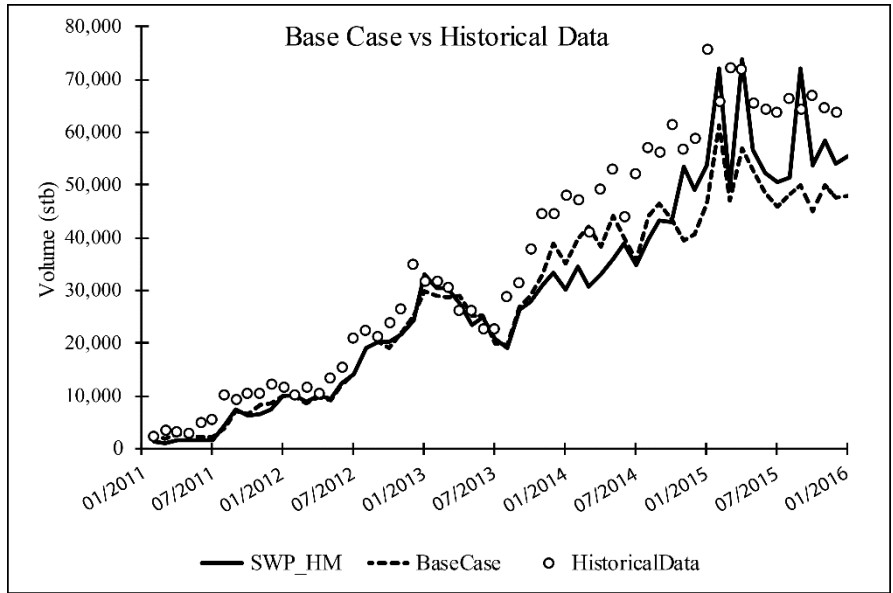

**Figure 4.** Plot shows the historical monthly oil production for the Farnsworth Units (black open circles), Base Case model forecasts (black dashed line), and history-matched model forecasts (SWP_HM) (solid black line). Both numerical models show a good data fit to the historical production data.

## 4. Discussion

Results indicate that the shape and endpoints of the relative permeability curve impart a significant influence on model forecasts. Generally, the more non-linear the curve, the more $CO_2$ that is predicted to be stored, and the less oil, water, and gas is predicted to be produced. However, the saturation endpoints and the assumed maximum relative permeability have a significant impact on this trend, with some of the more non-linear curves (C11 & C12) predicting more oil production than the most linear curve (C1).

### 4.1. Carbon Dioxide Storage

The total amount of stored $CO_2$ forecasted in this modeling study is between 2.8 million tons and 3.2 million tons by the end of the simulation. The largest fraction of the $CO_2$ is dissolved in the oil phase, between 1.2 and 2.4 million tons. The cases using highly non-linear relative permeability curves with narrow saturation ranges (C11 through

C16) had overall a larger portion of the $CO_2$ dissolved in the oil phase while the more linear curves with broader saturation ranges (C1 through C7) have almost a million tons less $CO_2$ dissolved in the oil phase but significantly more $CO_2$ in the supercritical phase. This indicates that the $CO_2$ in the supercritical phase has an inverse relationship to the $CO_2$ dissolved in the oil phase with respect to the changes in the relative permeability curve (Figure 5). As the curve becomes more non-linear, the fluid mobility decreases, leading to a decrease in the volume of $CO_2$ dissolved in the oil and a corresponding increase in the supercritical $CO_2$. The highest fluid mobility curve (model C1) predicts an almost even $CO_2$ distribution between the supercritical phase (46%) and oil phase (44%); while the lowest fluid mobility curves (C14 to C16) predict a significant difference in $CO_2$ phase distribution, 10% in the supercritical phase and 82% dissolved in the oil phase for model C14, see Table 3 for stored $CO_2$ mass distribution. The $CO_2$ mass dissolved in the water phase is mostly unaffected by changes in the relative permeability curve, varying by only 2% across all model permutations, from 8% to 10% of the total $CO_2$ stored. The Base Case (BC) model predicts the largest mass of $CO_2$ stored in the reservoir and the phase distribution matches the models with relative permeability curves that describe intermediate fluid mobility. This indicates that the Base Case relative permeability curve describes an intermediate fluid mobility relationship, close to what C8 and C9 models predict.

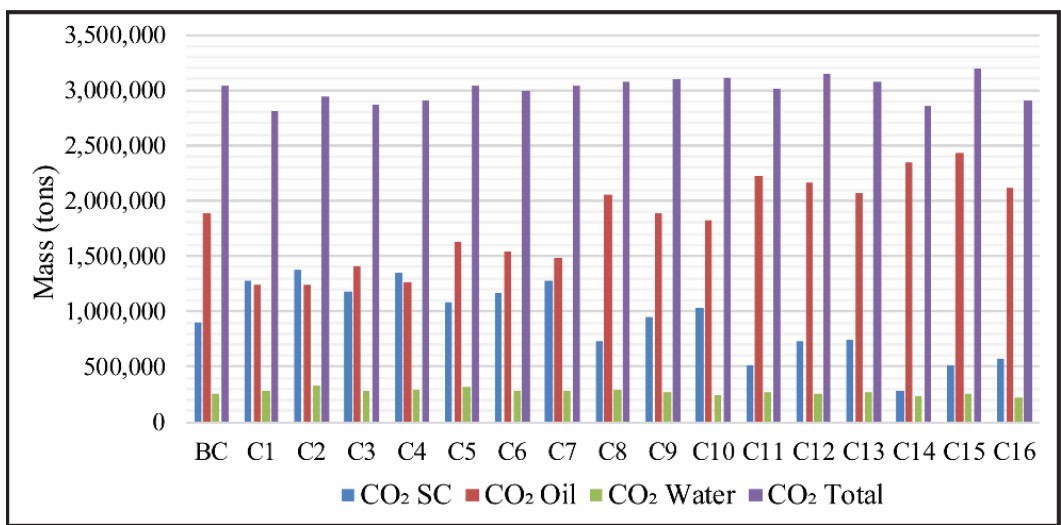

**Figure 5.** This chart shows $CO_2$ storage for each of the model permutations. '$CO_2$ SC' is the supercritical $CO_2$, '$CO_2$ Oil' is the $CO_2$ and $CH_4$ that is dissolved in the oil phase, '$CO_2$ Water' is the $CO_2$ dissolved in the aqueous phase, and '$CO_2$ Total' is the total amount of gas, both $CO_2$ and $CH_4$, in the reservoir.

**Table 3.** The total mass of $CO_2$ stored in each phase; supercritical, dissolved in the oil phase, dissolved in the aqueous phase. These data are the totals for the end of the simulation.

| Model | Supercritical $CO_2$ | | $CO_2$ in Oil | | $CO_2$ in Water | | Total |
|-------|----------------------|-----|---------------|-----|-----------------|-----|-------|
| | (Tons) | (%) | (Tons) | (%) | (Tons) | (%) | (Tons) |
| **BC** | $8.98 \times 10^5$ | 30% | $1.88 \times 10^6$ | 62% | $2.61 \times 10^5$ | 9% | $3.04 \times 10^6$ |
| **C1** | $1.28 \times 10^6$ | 46% | $1.24 \times 10^6$ | 44% | $2.79 \times 10^5$ | 10% | $2.80 \times 10^6$ |
| **C2** | $1.37 \times 10^6$ | 47% | $1.24 \times 10^6$ | 42% | $3.24 \times 10^5$ | 11% | $2.94 \times 10^6$ |
| **C3** | $1.18 \times 10^6$ | 41% | $1.41 \times 10^6$ | 49% | $2.80 \times 10^5$ | 10% | $2.87 \times 10^6$ |
| **C4** | $1.35 \times 10^6$ | 46% | $1.27 \times 10^6$ | 44% | $2.89 \times 10^5$ | 10% | $2.90 \times 10^6$ |
| **C5** | $1.08 \times 10^6$ | 36% | $1.63 \times 10^6$ | 54% | $3.18 \times 10^5$ | 10% | $3.03 \times 10^6$ |
| **C6** | $1.16 \times 10^6$ | 39% | $1.54 \times 10^6$ | 52% | $2.81 \times 10^5$ | 9% | $2.99 \times 10^6$ |
| **C7** | $1.28 \times 10^6$ | 42% | $1.48 \times 10^6$ | 49% | $2.77 \times 10^5$ | 9% | $3.04 \times 10^6$ |
| **C8** | $7.30 \times 10^5$ | 24% | $2.05 \times 10^6$ | 67% | $2.97 \times 10^5$ | 10% | $3.08 \times 10^6$ |
| **C9** | $9.46 \times 10^5$ | 31% | $1.89 \times 10^6$ | 61% | $2.66 \times 10^5$ | 9% | $3.10 \times 10^6$ |
| **C10** | $1.03 \times 10^6$ | 33% | $1.83 \times 10^6$ | 59% | $2.50 \times 10^5$ | 8% | $3.11 \times 10^6$ |
| **C11** | $5.14 \times 10^5$ | 17% | $2.22 \times 10^6$ | 74% | $2.75 \times 10^5$ | 9% | $3.01 \times 10^6$ |
| **C12** | $7.27 \times 10^5$ | 23% | $2.17 \times 10^6$ | 69% | $2.53 \times 10^5$ | 8% | $3.15 \times 10^6$ |
| **C13** | $7.48 \times 10^5$ | 24% | $2.06 \times 10^6$ | 67% | $2.69 \times 10^5$ | 9% | $3.08 \times 10^6$ |
| **C14** | $2.84 \times 10^5$ | 10% | $2.34 \times 10^6$ | 82% | $2.28 \times 10^5$ | 8% | $2.85 \times 10^6$ |
| **C15** | $5.08 \times 10^5$ | 16% | $2.43 \times 10^6$ | 76% | $2.52 \times 10^5$ | 8% | $3.19 \times 10^6$ |
| **C16** | $5.77 \times 10^5$ | 20% | $2.11 \times 10^6$ | 73% | $2.22 \times 10^5$ | 8% | $2.91 \times 10^6$ |

*4.2. Oil Production*

Oil production does not follow the same trend as the $CO_2$ storage. Models C1 through C9 indicate a declining oil production as fluid mobility described by the relative permeability curves decreases, except models C4 and C7 (Table 4). The relative permeability curve used in models C11, C12, and C14 describes a lower fluid mobility condition but predicts similar oil production as the highest fluid mobility models (C1 and C2); whereas the lowest fluid mobility relative permeability models predict the lowest oil production, as would be expected. A possible reason for the high oil production show in models C11, C12, and C14 is that the oil saturation falls within a range on the relative permeability curve that promotes the oil phase mobility, the mid-range of the relative permeability curve. Oil saturations are between 40% and 60% in the active production and injection areas giving relative permeability ranges of 0.23 to 0.35. Within the same area, the relative permeabilities to water never get above 0.1, promoting oil mobility over water mobility.

**Table 4.** Table shows the forecasted total oil produced for each simulation case and the magnitude of difference when compared to the Base Case (BC) forecasts.

| Model | Oil Produced (STB) | |
| --- | --- | --- |
| | Total | Delta vs. BC Model |
| **BC** | $2.16 \times 10^7$ | 0 |
| **C1** | $2.68 \times 10^7$ | $5.14 \times 10^6$ |
| **C2** | $2.57 \times 10^7$ | $4.08 \times 10^6$ |
| **C3** | $2.32 \times 10^7$ | $1.59 \times 10^6$ |
| **C4** | $2.77 \times 10^7$ | $6.03 \times 10^6$ |
| **C5** | $2.20 \times 10^7$ | $3.56 \times 10^5$ |
| **C6** | $2.04 \times 10^7$ | $-1.23 \times 10^6$ |
| **C7** | $2.60 \times 10^7$ | $4.39 \times 10^6$ |
| **C8** | $1.89 \times 10^7$ | $-2.77 \times 10^6$ |
| **C9** | $1.77 \times 10^7$ | $-3.96 \times 10^6$ |
| **C10** | $2.30 \times 10^7$ | $1.38 \times 10^6$ |
| **C11** | $2.63 \times 10^7$ | $4.69 \times 10^6$ |
| **C12** | $2.90 \times 10^7$ | $7.40 \times 10^6$ |
| **C13** | $2.26 \times 10^7$ | $9.84 \times 10^5$ |
| **C14** | $2.60 \times 10^7$ | $4.33 \times 10^6$ |
| **C15** | $1.51 \times 10^7$ | $-6.54 \times 10^6$ |
| **C16** | $1.75 \times 10^7$ | $-4.17 \times 10^6$ |

The Base Case (BC) model's oil production falls within the middle of the range predicted by this study, similar to models C6, C10, C13 (Table 4). This indicates that the Base Case relative permeability curve was measured from an area of the field that exhibits medium fluid mobility when compared to the range of fluid mobilities predicted by the synthetic relative permeability curves.

*4.3. Pressure*

The influence of relative permeability on the pressure field was highly time dependent. Figure 6 indicates that during the first phase of injection when historical data are used to control the injection rates (2010 to 2017), there is less difference in pressure between the relative permeability curves tested. During the predictive phase of the injection schedule (2017 to 2036), the difference across all the relative permeability curves tested increased to 27% by the end of injection. This variation in pressure increases significantly as the proportion of recycled $CO_2$ in the injection stream increases. An inflection point in all of the models on 1 January 2024 marks the point when $CO_2$ availability for injection becomes tied to production volumes and hence the fluid movement within the reservoir. On 1 January 2030, there is another inflection point that marks when new $CO_2$ to the model is stopped and only recycled $CO_2$ is available to the injection wells.

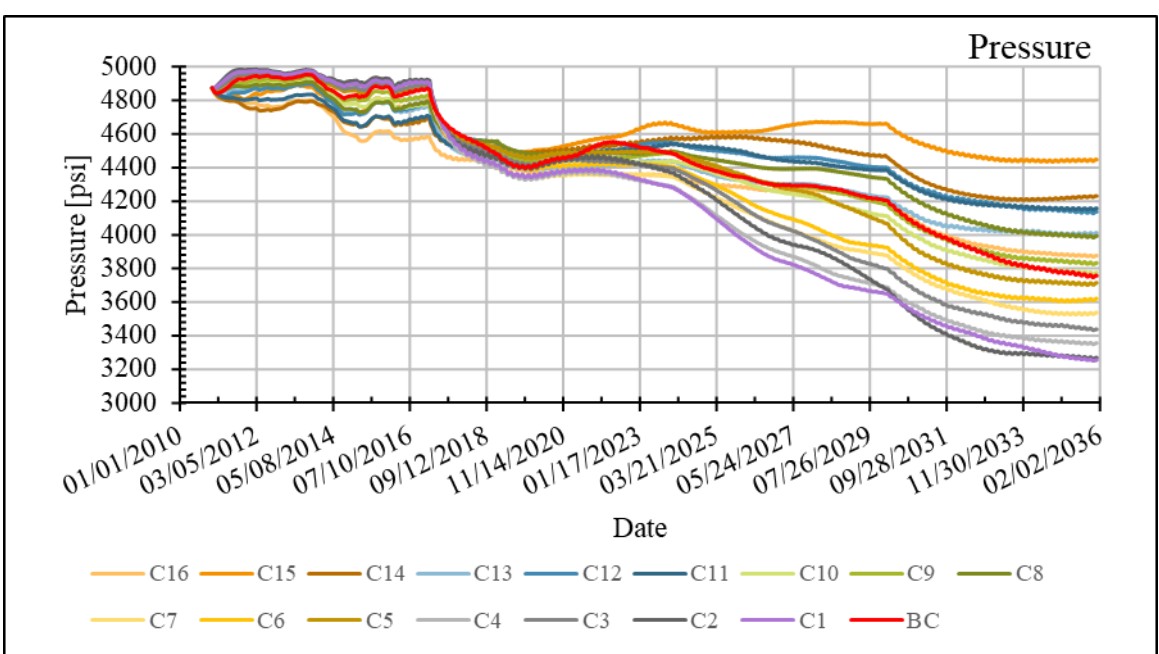

**Figure 6.** Pressure across the whole domain through time. The bold red line is the Base Case results.

The relative permeability curves appear to have a smaller, but still significant, impact on the average reservoir pressure when there is an unlimited source of $CO_2$ for the injection wells to meet their targets. Once the fluid mobility within the reservoir influences the volume of $CO_2$ available to meet injection targets, the impact of the relative permeability curves becomes much more pronounced. Curves that restrict the fluid movement through a high degree of curvature and narrow saturation range, such as C15 and C16, have lower oil production, less $CO_2$ present in the reservoir, and a higher average pressure, while the curves that promote fluid movement through more linear curvature and a wider saturation range (C1 and C2) have higher oil production, more $CO_2$ in the reservoir, and a lower average reservoir pressure.

## 5. Conclusions

Results of this study indicate that small variations in the shape of the relative permeability curve have a significant impact on the model forecasts. While all models predicted nearly the same total $CO_2$ stored in the reservoir, the phase it is stored as (supercritical vs. dissolved in oil vs. dissolved in water) is greatly influenced by the relative permeability curve. Relative permeability curves that describe low fluid mobility predict most of the $CO_2$ is stored in the oil phase with very little in the supercritical phase. The relative permeability curves that describe the highest overall fluid mobility predicts that there is an even distribution of $CO_2$ in the supercritical phase and the oil phase, allowing the $CO_2$ to migrate faster and in greater quantities to the production wells, leading to lower amounts of total stored $CO_2$. The higher the mobility, the more contact between the $CO_2$ plume and the oil, increasing the amount of $CO_2$ that is dissolved in the oil phase and thereby increasing production when compared to the relative permeability curves that predict low overall fluid mobility. The reduction in the oil's viscosity due to $CO_2$ dissolution allows it greater fluid mobility and may be why there is an increase in oil production when compared to some of the relative permeability curves that predict low fluid mobility. This may also account for why there is less $CO_2$ stored in the oil phase with the relative permeability curves that describe high fluid mobility.

The findings of this study indicate that the relative permeability curve is a critical parameter that must be given careful consideration when designing multiphase flow models. It is essential to understand and quantify the uncertainty in the relative permeability curve.

If measured laboratory relative permeability data are not available or limited for the study domain, the relative permeability curve should be considered a significant source of model uncertainty and accounted for as part of the simulation effort.

**Author Contributions:** Conceptualization, N.M. and B.M.; Data curation, N.M.; Formal analysis, N.M.; Funding acquisition, B.M.; Investigation, N.M.; Methodology, N.M., W.J. and B.M.; Project administration, B.M.; Software, N.M.; Supervision, B.M.; Validation, W.A.; Visualization, N.M.; Writing—original draft, N.M.; Writing—review & editing, N.M., W.A., W.J. and B.M. All authors have read and agreed to the published version of the manuscript.

**Funding:** Funding for this study was provided by the Southwest Regional Partnership on Carbon Sequestration (SWP) under Award No. DE-FC26-05NT42591.

**Institutional Review Board Statement:** Not applicable.

**Informed Consent Statement:** Not applicable.

**Conflicts of Interest:** The authors declare no conflict of interest. The funders had no role in the design of the study; in the collection, analyses, or interpretation of data; in the writing of the manuscript, or in the decision to publish the results.

**Appendix A**

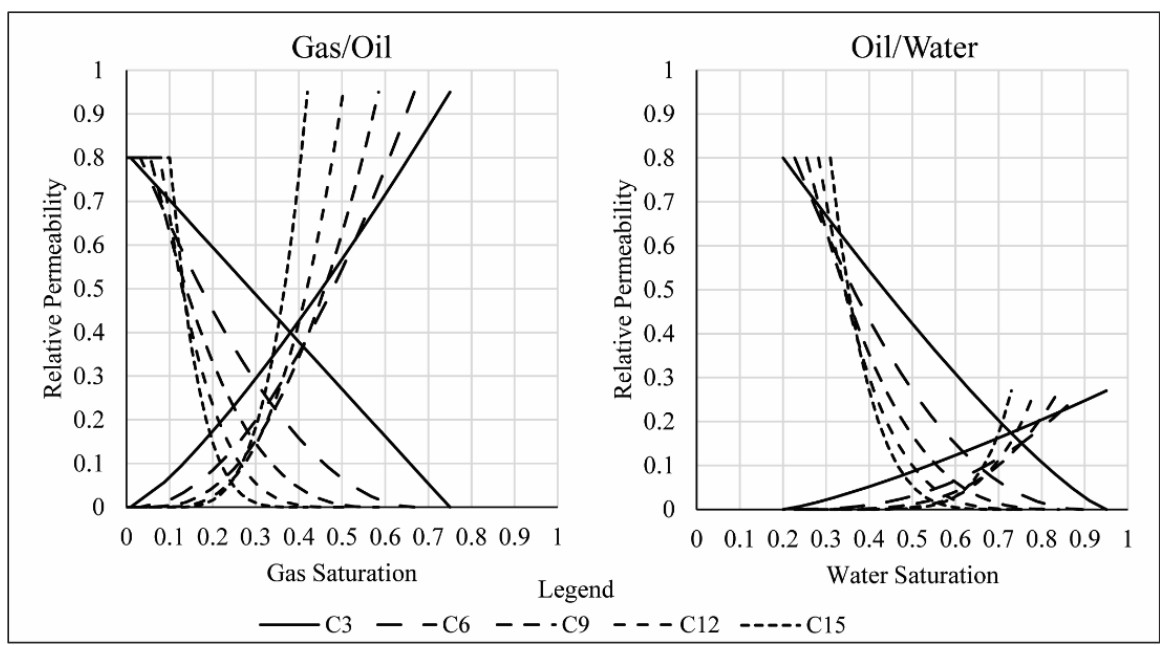

**Figure A1.** The C3, C6, C9, C12, and C15 relative permeability curves. The saturation endpoints and curvature are varied, while the relative permeability end points remain constant.

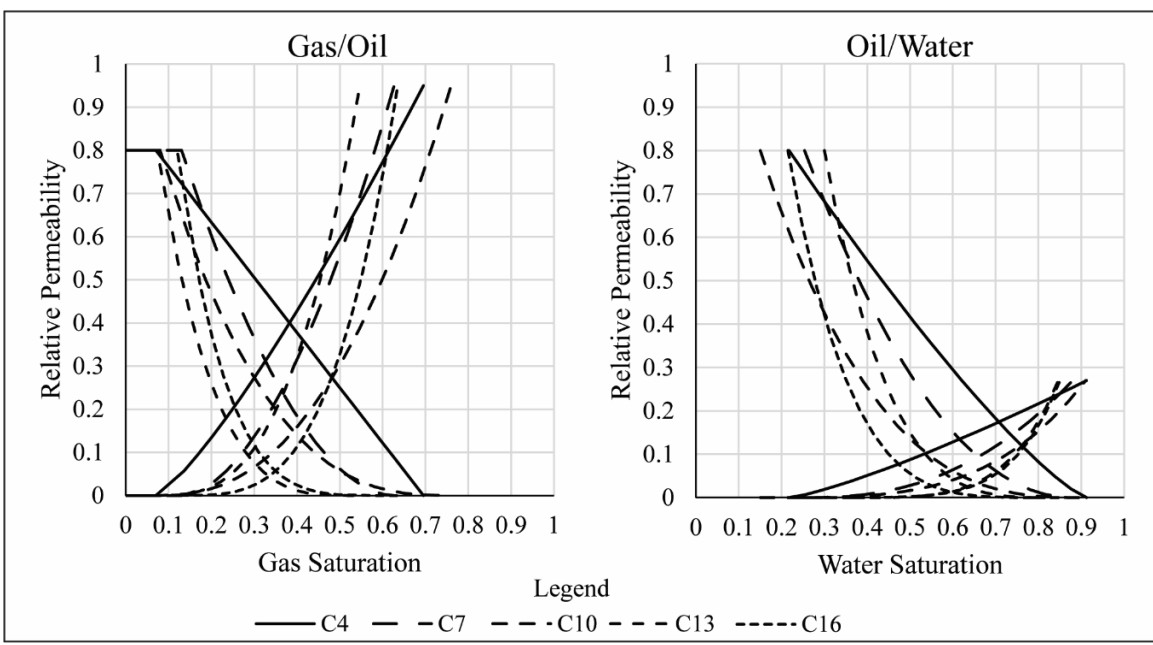

**Figure A2.** The C4, C7, C10, C13, and C16 relative permeability curves. The saturation endpoints and curvature are varied and the relative permeability endpoints are fixed.

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
