# Peer review of "Relative Permeability: A Critical Parameter in Numerical Simulations of Multiphase Flow in Porous Media"

_energies, doi:10.3390/en14092370_

Round 1

Reviewer 1 Report

Even if, topics addressed in the present work are meaningful and need to be widely deepened, this article seems to be poorly constructed and must be improved before being considered for possible publication.

First of all, parameters and subscript used in equation 1-4 and in the following tables, were not described adequately; in this way they are not so useful.

In the introduction, you wrote about two possible methods: the first pairs two-phase relative permeability, while the other calculates the three-phase relative permeability. Considering methodology and scopes of your research, why did not you evaluate the three-phase relative permeability and provided related diagrams?

In section 2, porosity was affirmed to range from 10 to 32%, however Figure 3 is in contrast with it. 

In addition, which method you followed to define the injection and the production wels?

In section dedicated to CO2 storage, you did not mention possible hydrate formation. Considering sandstones porosity and permeability, the injection pressure and the overall depth (7550-7950 feet), hydrate formation may occur (see Arabian Journal of Geosciences, 13 (2020) 898; Processes, 8 (2020) 1298), thus favouring CO2 storage or, on the other hand, hindering gas injection. The present possibility should be considered.

In Figure 6, a significant trend variation is present between 2018 and 2020, can you provide an explanation about it?

Title of section 5: Conclusions instead of Discussion

References should show the extended name of all authors (remove "et al.)

Reviewer 2 Report

  1. As the authors commented, relative permeability is critical for describing multiphase flow in porous media. Thus, it's of great theoretical and scientific significance to study the influence of relative permeability on multiphase flow. In this regard, I think it's an interesting work. However, the authors just applied different relative permeability curves based on Corey's model to conduct the research, which I think is not enough. I would like to recommend that the authors add more details about relative permeability and its influence on fluid flow. For example, besides Corey's model, there are also some other models. which is more suitable for describing the flow problem in porous media.
  2. I think the word "is" in line 19 should be deleted.
  3. I think the subtitle of part 5 is "conclusions" not "Discussion"
  4. What's the innovation of this work? Please explain it.

Round 2

Reviewer 1 Report

The paper can now be considered for publication